REGISTERED REPORT PROTOCOL

# HIV-1 Gag gene mutations, treatment response and drug resistance to protease inhibitors: A systematic review and meta-analysis protocol

**Alex Durand Nka**[1,2,3], **Georges Teto**[1☯*], **Maria Mercedes Santoro**[2☯], **Valantine Ngum Ndze**[4☯], **Désiré Takou**[1], **Beatrice Dambaya**[1], **Ezechiel Ngoufack Jagni Semengue**[1,2,3‡], **Lavinia Fabeni**[5], **Carlo-Federico Perno**[6], **Vittorio Colizzi**[1,2,3,7], **Francesca Ceccherini-Silberstein**[2], **Joseph Fokam**[1,4‡*]

1 Chantal BIYA International Reference Centre for research on HIV/AIDS Prevention and Management, (CIRCB), Yaoundé, Cameroon, 2 University of Rome "Tor Vergata", Rome, Italy, 3 Evangelical University of Cameroon, Bandjoun, Cameroon, 4 Faculty of Medicine and Biomedical Sciences, University of Yaoundé I, Yaoundé, Cameroon, 5 Laboratory of Virology, National Institute for Infectious Diseases "Lazzaro Spallanzani" -IRCCS, Rome, Italy, 6 IRCCS Bambino Gesu Children's Hospital, Rome, Italy, 7 Chair of Biotechnology-UNESCO, University of Rome "Tor Vergata", Rome, Italy

☯ These authors contributed equally to this work.
‡ These authors also contributed equally to this work.
* josephfokam@gmail.com (JF); ggeto@yahoo.fr (GT)

This is a Registered Report and may have an associated publication; please check the article page on the journal site for any related articles.

## Abstract

### Background

Some mutations in the HIV-1 *Gag* gene are known to confer resistance to ritonavir-boosted protease inhibitors (PI/r), but their clinical implications remain controversial. This review aims at summarizing current knowledge on HIV-1 Gag gene mutations that are selected under PI/r pressure and their distribution according to viral subtypes.

### Materials and methods

Randomized and non-randomized trials, cohort and cross-sectional studies evaluating HIV-1 Gag gene mutations and protease resistance associated mutations, will all be included. Searches will be conducted (from January 2000 onwards) in PubMed, Embase, Cochrane Central Register of Controlled Trials (CENTRAL), Latin American and Caribbean Health Sciences Literature (LILAC), Web of Science, African Journals Online, and Cumulative Index to Nursing and Allied Health Literature (CINAHL) databases. Hand searching of the reference lists of relevant reviews and trials will be conducted and we will also look for conference abstracts. Genotypic profiles of both Gag gene and the protease region as well as viral subtypes (especially B vs. non B) will all serve as comparators. Primary outcomes will be the "prevalence of Gag mutations" and the "prevalence of PI/r resistance associated mutations". Secondary outcomes will be the "rate of treatment failure" and the distribution of Gag mutations according to subtypes. Two reviewers will independently screen titles and abstracts, assess the full texts for eligibility, and extract data. If data permits, random effects

**Data Availability Statement:** All relevant data are within the manuscript and its Supporting Information files.

**Funding:** The authors received no specific funding for this work.

**Competing interests:** No funding or sponsorship was received for this study protocol or publication of this article.

models will be used where appropriate. This study will be reported according to the guidelines of the Preferred Reporting Items for Systematic Reviews and Meta Analyses.

## Discussion

This systematic review will help identify HIV-1 Gag gene mutations associated to PI/r-based regimen according to viral subtypes. Findings of this review will help to better understand the implications of the Gag gene mutations in PI/r treatment failure. This may later justify considerations of Gag-genotyping within HIV drug resistance interpretation algorithms in the clinical management of patients receiving PI/r regimens.

## Systematic review registration

PROSPERO: CRD42019114851.

## Background

The main cause of HIV-1 antiretroviral treatment (ART) failure is drug resistance development [1–3]. In resource limited settings (RLS), despite the transition to Dolutegravir (third line), there are three major recommended ART lines: non-nucleoside reverse transcriptase inhibitors (NNRTI) based first line and boosted protease inhibitors (PI/r) based second-line, each with two nucleoside reverse transcriptase inhibitors (NRTI) [4]. However, already observed second-line failure is estimated to escalate with limited third line options [4]. Detection and understanding second line drug résistance is therefore important to sustain treatment efficacy and plan for effective future therapy in resource limited setting.

Known PI/r mechanisms involve the development of mutations in and around the protease active site that change its interaction with the inhibitors [4, 5]. Such changes typically lead to fitness costs and loss of replication capacity that can be compensated by more distant protease mutations [6]. However, whereas >70% of patients who fail first/second-line failure have reverse transcriptase (RT) drug resistance mutations (DRMS), patients failing second-line ART have low levels (average 18%) of PI/r drug resistance mutations, which is not well understood [7]. This observation suggests alternative drug resistance mechanisms that lead to PI/r treatment failure, such as non-adherence or decreased PI levels [8, 9], resistant minor variants not detected by conventional assays [10], or region outside the protease such as Gag gene [11].

The amino acid changes in the viral enzyme, a unique feature of HIV resistance to protease inhibitors is that resistance mutations occur not only in the protease itself- the direct target of the inhibitors, but also is one natural substrates of the protease, the Gag gene. These mutations enhance the interactions between the substrate and the mutated enzyme and increase the cleavage accordingly. Initially described as compensatory mutations capable of partially correcting the loss of viral capacity resulting from protease mutations, Gag mutations are now recognized as being directly involved in resistance [12, 13].

The current review aims to examine the evidence that HIV-Gag gene mutations are an important component of HIV resistance to PI/r by addressing in detail mutations in Gag gene that are selected under PI/r pressure and how this mutations vary (according to viral subtype). The finding of this review may help to better understand the implications of the Gag gene mutations in PI/r treatment failure, for potential considerations in algorithms interpreting HIV drug resistance for clinical management.

## Materials and method

### Design and registration

This systematic review and meta-analysis protocol has been conducted following the recommendation of the Preferred Reporting Items for Systematic Reviews and Meta-Analysis (PRISMA-P) guidelines [14]. This review protocol is registered in the Prospective Register of systematic Reviews (PROSPERO) system and can be accessed at https://www.crd.york.ac.uk/prospero/displayrecord.php?ID=CRD42019114851. See (S1 Checklist) for the completed PRISMA-P checklist.

### Design and setting of the study

**Inclusion criteria.**   *1. Type of studies*. We will include randomized and non-randomized trials, cohort and cross-sectional studies evaluating HIV-1 Gag gene mutations and protease resistance associated mutations.

*2. Type of participants*. We will consider studies conducted among HIV-infected individuals without distinction of age, sex and geographical location.

*3. Intervention*. PI/r-based regimens will be our intervention of interest. Studies focusing on patients under PI/r that are approved by the Food and Drug Administration (FDA) (Atazanavir (ATV/r), Darunavir (DVR/r), Lopinavir (LPV/r), Fosamprenavir (FPV/r), Indinavir (IDV/r), Nelfinavir (NVF), Saquinavir (SQV/r) and Tiprananvir (TPV/r)) based regimens will be considered as our first group of interest. Our second group of interest will be made up of studies on patients under PI-sparing regimens. Finally, studies on naïve patients will serve as control group.

*4. Comparators*. Given that studies onHIV-1 Gag gene mutations and protease resistance associated mutations will be the only ones included, genotypic profiles of both the Gag gene and the protease region will serve as comparators. Also, given the wide genetic variability of HIV-1, viral subtypes (and especially B vs. non B) may serve as comparators as well.

*5. Types of outcomes*. Primary outcomes will be the "prevalence of Gag mutations" and the "prevalence of PI/r resistance associated mutations". Secondary outcomes will be the "rate of treatment failure" and the distribution of Gag mutations according to subtypes. HIV-1 Gag gene mutations will be considered as non-polymorphic mutations (< 5% occurrence) according to gene portion (cleavage and non-cleavage sites). The reference sequence that will be used to determine Gag gene mutations will be the Human immunodeficiency virus type 1 (HXB2) reference genome. PI/r resistance associated mutations will be considered as defined by Stanford HIVdb list (https://hivdb.stanford.edu/hivdb/by-mutations/) and the 2019-International AIDS Society drug resistance list [15]. As there are discrepancies between the Stanford mutation list and the IAS drug resistance list, we will consider the IAS list as the last resort. We will compare the emergence of Gag mutations according to HIV viral load, and treatment failure ($\geq$1000 copies/ml) [16] will not be a criteria for inclusion.

*6. Report characteristics*. We will include studies that have been published from January 2000 onwards without any restriction of language.

**Exclusion criteria.**   Case reports, letters, comments, editorials and case series (<30 participants).

### Search strategy

A comprehensive search will be conducted in the major databases-PubMed/MEDLINE, Excerpta Medica database (EMBASE), Cochrane Central Register of Controlled Trials

(CENTRAL), and Cumulative Index to Nursing and Allied Health Literature (CINAHL)-using key terms and Medical Subject Heading (MeSH) specifically designed for the respective databases. The Medical Subject Headings (MeSH) [14] for HIV and key words "Gag mutations", "Protease inhibitors", "drug resistance", "Subtypes" 'will be cross-referenced (S1 File shows the detailed search strategy for Pubmed, Embase and CINAHL). We will update the search prior to publication to include any additional eligible papers published recently. Ongoing trials will be sought in the WHO International Clinical Trials Registry Platform (ICTRP) and Clinical-Trials.gov. We will search conference abstract archives on the websites of the Conference on Retroviruses and Opportunistic Infections (CROI); the International AIDS Conference (IAC); the International AIDS Society Conference on HIV Pathogenesis, Treatment, and Prevention (IAS) and all Virology Education conferences, for all available abstracts presented at all conferences from January 2000 onwards. Hand searching of the reference lists of relevant reviews and trials will be conducted. In addition, we will contact experts in the field for other potentially eligible studies we may have missed.

## Data management

All records from the various sources included in our search strategy will be combined, uploaded into the reference management software Zotero® (version 5.0.85) and de-duplicated.

## Selection of studies for inclusion in the review

Articles retrieved from databases will be independently selected by two authors (AND, ENJS) using"SysRev" (https://sysrev.com/), a software for a systematic review production tool for title/abstract screening. Any disagreement will be solved by discussion and consensus, or will involve a third review author (GT) as an arbitrator. Studies written in languages different from English or French will be translated using Deepl Translator® and considered for eligibility. Two review authors (AND, GT) will independently evaluate the full text of the selected records. Discrepancies will be resolved by consensus or by an arbitration of a third review author (MMS). The agreement between the two first review authors will be estimated by Cohen's kappa coefficient.

## Data extraction and management

We will extract the following from included studies:

- Study characteristics (name of the first author, year of publication, study period, study design, aim of the study, country in which the study was conducted)

- Characteristics of study population (sample size, inclusion and exclusion criteria, duration of follow-up)

- ART-exposure

- PI/r based regimens

- HIV-1 viral subtype

- Gag gene mutations

- Protease resistance associated mutations

- Viral load

HIV-1 Gag mutations will be classified according to Gag gene portion (cleavages and non-cleavages sites) and the exposure of PI/r based regimen. Disagreements between the two review authors will be solved by discussion, or if necessary will involve a third review author.

## Data synthesis

Data will be analyzed using the 'meta' and 'metafor' packages of the R statistical software (V.3.4.4, R Foundation for Statistical Computing, Vienna, Austria).A descriptive analysis of study characteristics will be undertaken to explore the heterogeneity of the studies. Summary statistics will then be used to describe study outcomes, including means or medians, and frequencies. Proportions with exact binomial 95% confidence intervals (95% CI) will be calculated for each outcome and presented in forest plots. Heterogeneity will be evaluated by the $\chi^2$ test on Cochran's Q statistic, which will be quantified by H and $I^2$ values. The $I^2$ statistic estimates the percentage of total variation across studies due to true between-study differences rather than chance. In general, $I^2$ values greater than 60%–70% indicate the presence of substantial heterogeneity. In the case of substantial heterogeneity, subgroup and meta-regression analyses will be used to investigate sources of heterogeneity. Subgroup analyses will be performed for the following subgroups: HIV-1 Gag non polymorphic mutations according to gene portion (cleavage and non-cleavage sites), viral subtypes (B VS non B subtypes), protease inhibitors regimen, age (pediatrics/adolescents and adults), and country. Univariable and multivariable meta-regression will be used to test for an effect of PI/r regimen and viral subtype on Gag gene non polymorphic mutations. A p value <0.05 will be considered statistically significant. To be included in multivariable meta-regression analysis, a p value <0.25 in univariable analysis will be required. We will use the GRADE approach to rate the certainty of evidences as "high", "moderate", "low" and "very low (S2 File).

## Risk of bias assessment

The evaluation of included studies for risk bias will be done using ROBINS-1 [17, 18], a tool for assessing risk of bias in non-randomized studies of interventions. ROBIS [RoB 2.0] [19, 20] will be used for randomized controlled trials studies. Discrepancy in risk of bias assessment among the review authors will be solved by discussion and consensus, or by arbitration of a third review author.

# Discussion

This study will help to better understand protease inhibitors treatment failure in HIV-infected individuals by highlighting the involvement of Gag gene mutations in this resistance process. The study will also shed light on the variation of mutations in the Gag gene as a function of the distribution of the different subtypes of HIV-1. Our results will be useful to virologists and physicians in their understanding of the resistance to protease inhibitors without resistance mutations in the protease gene; which may provide important arguments for a possible introduction of the Gag portion into the HIV resistance-testing panel. As potential limitations of this review, we may be confronted with important study heterogeneity and incompleteness, but these will be considered in statistic models during meta-regression analysis; if not perform, studies incompleteness at least would be solve by contacting study authors. Another limitation may be at the level of reviewing and including studies. In effect, in the process of resolving disagreements while reviewing articles, all team members will be included in the decision-making process or at least aware of the disagreements being discussed. We will try as much as possible to have a consensus decision for each disagreement. Important protocol amendments will be documented, taken into consideration while analyzing the data, and discussed consequently in

the final paper. Our findings will be published in a peer-review journal and subsequently disseminated to policy-makers first at the national level through the submission of a governmental notice, and at the international level through conferences and stakeholder meetings.

## Supporting information

**S1 Checklist. PRISMA-P 2015 checklist.**
(DOCX)

**S1 File. Search strategy.**
(DOCX)

**S2 File. Assessing the quality of evidences and the strength of recommendations.**
(DOCX)

## Acknowledgments

The authors would like to thank the "Chantal BIYA" International Reference Centre to hosting the Work sessions of drafting the Systematic Review protocol and all the authors with the work already done on Gag studies.

## Author Contributions

**Conceptualization:** Alex Durand Nka, Valantine Ngum Ndze, Désiré Takou, Francesca Ceccherini-Silberstein, Joseph Fokam.

**Formal analysis:** Alex Durand Nka, Valantine Ngum Ndze, Lavinia Fabeni.

**Methodology:** Alex Durand Nka, Désiré Takou, Beatrice Dambaya, Lavinia Fabeni.

**Supervision:** Georges Teto, Maria Mercedes Santoro, Francesca Ceccherini-Silberstein, Joseph Fokam.

**Validation:** Georges Teto, Maria Mercedes Santoro, Beatrice Dambaya, Carlo-Federico Perno, Vittorio Colizzi, Francesca Ceccherini-Silberstein, Joseph Fokam.

**Writing – original draft:** Alex Durand Nka, Georges Teto, Ezechiel Ngoufack Jagni Semengue, Joseph Fokam.

**Writing – review & editing:** Alex Durand Nka.

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
