## [Decision Letter · Decision Letter 0]

5 May 2021

PONE-D-20-31760

HIV-1 Gag gene mutations, treatment response and drug resistance to Protease inhibitors: A systematic review and meta-analysis protocol

PLOS ONE

Dear Dr. NKA,

Thank you for submitting your manuscript to PLOS ONE. After careful consideration, we feel that it has merit but does not fully meet PLOS ONE’s publication criteria as it currently stands. Therefore, we invite you to submit a revised version of the manuscript that addresses the points raised during the review process.

I apologize for long delay in receiving reviews for your protocol. We have received 3 expert reviews at this time. As you can see, one asks for major revision (reviewer #1), while the other two reviewers have less concern and recommend minor revision of the protocol. I think reviewer #1 did not buy in to the registered report protocol format, so I am placing more weight on the comments of reviewer #2 and reviewer #3. Please do your best to respond to all three reviewers, and modify where it makes sense according to the suggestions. I think we can promise a more swift turnaround for the revised version.

We look forward to receiving your revised manuscript.

Kind regards,

Paul Spearman

Academic Editor

PLOS ONE

Journal Requirements:

5. Please include captions for your Supporting Information files at the end of your manuscript, and update any in-text citations to match accordingly. Please see our Supporting Information guidelines for more information: http://journals.plos.org/plosone/s/supporting-information

Additional Editor Comments:

Dear Dr. Durand:

I apologize for long delay in receiving reviews for your protocol. We have received 3 expert reviews at this time. As you can see, one asks for major revision (reviewer #1), while the other two reviewers have less concern and recommend minor revision of the protocol. I think reviewer #1 did not buy in to the registered report protocol format, so I am placing more weight on the comments of reviewer #2 and reviewer #3. Please do your best to respond to all three reviewers, and modify where it makes sense according to the suggestions. I think we can promise a more swift turnaround for the revised version.

Paul Spearman

Reviewers' comments:

Reviewer's Responses to Questions

**Comments to the Author**

1. Does the manuscript provide a valid rationale for the proposed study, with clearly identified and justified research questions?

Reviewer #1: No

Reviewer #2: Yes

Reviewer #3: Yes

2. Is the protocol technically sound and planned in a manner that will lead to a meaningful outcome and allow testing the stated hypotheses?

Reviewer #1: Partly

Reviewer #2: Yes

Reviewer #3: Yes

3. Is the methodology feasible and described in sufficient detail to allow the work to be replicable?

Reviewer #1: No

Reviewer #2: Yes

Reviewer #3: Yes

4. Have the authors described where all data underlying the findings will be made available when the study is complete?

Reviewer #1: Yes

Reviewer #2: No

Reviewer #3: Yes

5. Is the manuscript presented in an intelligible fashion and written in standard English?

Reviewer #1: Yes

Reviewer #2: Yes

Reviewer #3: Yes

6. Review Comments to the Author

You may also provide optional suggestions and comments to authors that they might find helpful in planning their study.

Reviewer #1: It is not clear what the aim of the protocol is for. The lack of a purpose of the study also makes it confusing to what unique purpose the review will serve given that there are already existing reviews looking at Gag mutations towards PI resistance, The biggest problem and lack is actually that there is lack of Gag mutations being mapped to specific PI resistance and the corresponding mutations in protease for compensatory mechanism.

The above needs to be addressed for a proper review.

Minor are typos in line 45, 125, 131 where words are just stuck together.

The Background or introduction should state clearly the point of the outcome of the work.

Some suggested points would be to investigate the mechanisms of the mutations.

Reviewer #2: The role of gag mutations/polymorphisms in resistance to PIs is still not clearly understood. This protocol outlines how the authors plan to investigate this by performing a meta-analysis on data (drug resistance, virological failure, subtypes) obtained from published papers and accessible databases on patients failing PI-based antiretroviral treatment. The main aim of this is to justify the inclusion of gag sequences in HIV-1 drug interpretation algorithms for managing patients on antiretroviral treatment. Although patients from all ages will be included, I do think that pediatric and adolescent and adult patients are unique sub-populations that need to be analyzed separately. That is also true for patients from low-to-middle income countries (vs. first-world countries). Although the authors will ultimately look at drug resistance mutations per se, which should not be impacted by age and country, there may be different adherence patterns (and consequently different levels of drug pressures) which may impact on the types of drug resistance mutations that are selected for (in PR and/or in gag). In addition, genotypic drug resistance data in adults may be scarcer than the data on paediatrics/adolescents, especially in low-to-middle income countries since PI-based regimens form the first-line regimen for pediatrics in these settings.

Some minor comments:

1. Please check typing as there are several instances of spaces missing between words.

2. In the Questions section under "Data Availability", please specify where the data will be made available.

3. In the Questions section under "Describe where the data may be found in full sentences: I would like to recommend that the website addresses of databases and search engines employed to obtain the data be added.

Reviewer #3: This systematic review protocol describes how the authors will set about collating information to identify HIV-1 Gag mutation associated with ritonavir-boosted protease inhibitors, according to viral subtypes. This is a very worthwhile study and the results should be very intriguing.

Some issues require further clarity:

1. Intervention – the first group will include patients receiving atazanavir, lopinavir and darunavir. Given the data search will extend to the year 2000, and that other PI have been used since that time, including nelfinavir, will patients receiving only the three PI’s mentioned, will other PI-based regimens be an exclusion criterion? Later in the protocol, indinavir is also mentioned.

2. Prevalence of gag mutations is defined as non-polymorphic mutations (<5%). However, as Gag mutations are not specifically defined, these mutations need to be relative to a reference sequence. Please clarify what will be used as the reference sequence.

3. As there are discrepancies between the Stanford mutation list and the IAS drug resistance list, please clarify which one will be used.

4. Treatment failure is defined as having an unsuppressed viral load (≥1,000 copies/ml). As treatment failure is inconsistently defined across different regions, and a first viraemic result sometimes considered as non-adherence, how will the analysis accommodate these differences. If any VL result ≥1000 copies/ml considered for inclusion, please clarify in the protocol.

5. Line 204 refers to duration of follow up. Is the absence of this data considered an exclusion criterion, as I predict this will be absent from a number of studies.

6. Line 91 suggest rephrasing this sentence to clarify that one gene is not “some”.

7. PLOS authors have the option to publish the peer review history of their article (what does this mean?). If published, this will include your full peer review and any attached files.

Reviewer #1: No

Reviewer #2: No

Reviewer #3: No

---

## [Author Response · Author response to Decision Letter 0]

21 May 2021

May 17th 2021

To the Editor-In-Chief,

PLOS ONE

Submission of revised manuscript: PONE-D-20-31760

Dear Editor-In-Chief:

Thank you for your email dated May 05 2021 enclosing yours and reviewers’ comments. We have carefully reviewed the comments and have revised the manuscript accordingly. Our responses are given in a point-by-point manner below. Changes to the manuscript are shown bold in this document and highlighted in yellow in the revised manuscript.

We hope the revised version is now suitable for publication and look forward to hearing from you in due course.

Cordially yours,

Dr Joseph Fokam / Dr Georges Teto

The Corresponding authors, on behalf of the co-authors

 

Title: HIV-1 Gag gene mutations, treatment response and drug resistance to protease inhibitors: A systematic review and meta-analysis protocol

Journal Requirements:

.

Answer: Thank you for this comment, we have indeed adjusted the manuscript according to PLOS ONE’S requirements: (page 1 line 27-31).

Answer: Thank you for this comment, we reviewed the list of references and the updated ones have been highlighted in yellow. Reference number 14 has been retracted, so we replaced it with a more recent one.

Important: If there are ethical or legal restrictions to sharing your data publicly, please explain these restrictions in detail. Please see our guidelines for more information on what we consider unacceptable restrictions to publicly sharing data:http://journals.plos.org/plosone/s/data-availability#loc-unacceptable-data-access-restrictions. Note that it is not acceptable for the authors to be the sole named individuals responsible for ensuring data access.

Answer: Thank you for this comment, our minimal underlying dataset was mentioned as a supporting information (additional file 1, 2 and 3) (Page 10 line 337, 339, and 341).

Answer: Thank you for this comment; My ORCID ID was already authenticate in Editorial Manager System.

5. Please include captions for your Supporting Information files at the end of your manuscript, and update any in-text citations to match accordingly. Please see our Supporting Information guidelines for more information: http://journals.plos.org/plosone/s/supporting-information

Answer: Thank you for this comment; Captions for the supporting information was included at the end of the Manuscript.

Additional Editor Comments:

Dear Dr. Durand:

I apologize for long delay in receiving reviews for your protocol. We have received 3 expert reviews at this time. As you can see, one asks for major revision (reviewer #1), while the other two reviewers have less concern and recommend minor revision of the protocol. I think reviewer #1 did not buy in to the registered report protocol format, so I am placing more weight on the comments of reviewer #2 and reviewer #3. Please do your best to respond to all three reviewers, and modify where it makes sense according to the suggestions. I think we can promise a more swift turnaround for the revised version.

Answer: Thank you for this comment; we did our best to address the concerns of all three reviewers.

Review Comments to the Author:

Reviewer #1: It is not clear what the aim of the protocol is for. The lack of a purpose of the study also makes it confusing to what unique purpose the review will serve given that there are already existing reviews looking at Gag mutations towards PI resistance, The biggest problem and lack is actually that there is lack of Gag mutations being mapped to specific PI resistance and the corresponding mutations in protease for compensatory mechanism.

The above needs to be addressed for a proper review.

Minor are typos in line 45, 125, 131 where words are just stuck together.

The Background or introduction should state clearly the point of the outcome of the work.

Some suggested points would be to investigate the mechanisms of the mutations.

Answer: Thank you for your comments.

Indeed, we have come across a number of reviews of mutations in the Gag gene and their likely implications for resistance to protease inhibitors. However, these reviews had the peculiarity that they did not characterize HIV-1 Gag gene mutations according to viral subtypes, and for some of them, they were only carried out in B subtypes. Therefore, our study would like to take into account the genetic diversity in the emergence of these mutations and their selection according to the Protease inhibitors regimen.

We agree with you regarding an investigation of the Gag gene mutations in the resistance of specific PI/r and the corresponding mutations in the protease for a compensatory mechanism. As part of our investigation we will classify these mutations according to the PI/r regimen which will give an idea of which Gag gene mutations are prioritized by certain PI/r

The minor are typos in line 45, 125 and 131 has been taken into account and we have already made corrections (page 2 line 45, page 4 line 127 and 133).

Reviewer #2: The role of gag mutations/polymorphisms in resistance to PIs is still not clearly understood. This protocol outlines how the authors plan to investigate this by performing a meta-analysis on data (drug resistance, virological failure, subtypes) obtained from published papers and accessible databases on patients failing PI-based antiretroviral treatment. The main aim of this is to justify the inclusion of gag sequences in HIV-1 drug interpretation algorithms for managing patients on antiretroviral treatment. Although patients from all ages will be included, I do think that pediatric and adolescent and adult patients are unique sub-populations that need to be analyzed separately. That is also true for patients from low-to-middle income countries (vs. first-world countries). Although the authors will ultimately look at drug resistance mutations per se, which should not be impacted by age and country, there may be different adherence patterns (and consequently different levels of drug pressures) which may impact on the types of drug resistance mutations that are selected for (in PR and/or in gag). In addition, genotypic drug resistance data in adults may be scarcer than the data on paediatrics/adolescents, especially in low-to-middle income countries since PI-based regimens form the first-line regimen for pediatrics in these settings.

Answer: Thank you for your comments.

We fully agree with you and this suggestion has been taken into account in the protocol (Page 7 line 209 -210). As you mentioned, the pediatric population is primarily exposed to PI/r in resource-limited countries, which will also allow a better appreciation of the emergence of Gag gene mutations according to age groups which would be influenced by the treatment strategies in each age group.

Some minor comments:

1. Please check typing as there are several instances of spaces missing between words.

Answer: Thank you for your comments. This was already done.

2. In the Questions section under "Data Availability", please specify where the data will be made available.

Answer: Thank you for this comment, this was already specify in this section.

3. In the Questions section under "Describe where the data may be found in full sentences: I would like to recommend that the website addresses of databases and search engines employed to obtain the data be added.

Answer: Thank you for this comment, this was already Describe in this section.

Reviewer #3: This systematic review protocol describes how the authors will set about collating information to identify HIV-1 Gag mutation associated with ritonavir-boosted protease inhibitors, according to viral subtypes. This is a very worthwhile study and the results should be very intriguing.

Some issues require further clarity:

1. Intervention – the first group will include patients receiving atazanavir, lopinavir and darunavir. Given the data search will extend to the year 2000, and that other PI have been used since that time, including nelfinavir, will patients receiving only the three PI’s mentioned, will other PI-based regimens be an exclusion criterion? Later in the protocol, indinavir is also mentioned.

Answer: Thank you for this comment, indeed, considering only participants who have been under Atazanavir (ATV/r), Lopinavir (LPV/r), and Darunavir (DRV/r) would limit the number of studies to be included in our systematic review, and its molecules are mostly taken in resource-limited countries. Therefore, we have adjusted the protocol and just specify that it will be patients under PI/r approved by the food drug administration (Page 4, line 120, 121, and 122).

2. Prevalence of gag mutations is defined as non-polymorphic mutations (<5%). However, as Gag mutations are not specifically defined, these mutations need to be relative to a reference sequence. Please clarify what will be used as the reference sequence

Answer: Thank you for this comment, the reference sequence that will be used to determine Gag gene mutations will be the Human immunodeficiency virus type 1 (HXB2) reference genome. This was already specify in the protocol (page 4, line 134,135,136).

3. As there are discrepancies between the Stanford mutation list and the IAS drug resistance list, please clarify which one will be used.

Answer: Thank you for this comment, In fact protease mutations will be firstly characterize according to Stanford HIVdb list and confirm by 2019-IAS list. As there are discrepancies between the Stanford mutation list and IAS drug resistance list, we will consider the IAS list as the last resort (Page 5, line 138 to 140).

4. Treatment failure is defined as having an unsuppressed viral load (≥1,000 copies/ml). As treatment failure is inconsistently defined across different regions, and a first viraemic result sometimes considered as non-adherence, how will the analysis accommodate these differences. If any VL result ≥1000 copies/ml considered for inclusion, please clarify in the protocol.

Answer: Thank you for this comment, the definition of virological failure depends on the context and in some countries it is a detectable viral load. Given the disparity of opinions about the definition of virological failure, we will just compare the emergence of HIV Gag mutations according to the viremia without mentioning whether it is a failure or not. And viremia status will not be a criterion for inclusion.

5. Line 204 refers to duration of follow up. Is the absence of this data considered an exclusion criterion, as I predict this will be absent from a number of studies.

Answer: Thank you for this comment, we have indeed removed this variable in order to be able to enroll as many studies as possible (line 210).

6. Line 91 suggest rephrasing this sentence to clarify that one gene is not “some”.

Answer: Thank you for this comment, this was already correct (line 91).

Once more, we are very appreciative for the valuable contributions made by the reviewers and the editor for improving the quality of our manuscript. We are also convinced that these comments were very necessary and we hope that the paper would now be deemed acceptable for publication in PloS One.

Dr Joseph Fokam and Georges Teto

The corresponding authors, on behalf of the coauthors

---

## [Editor Report · Decision Letter 1]

9 Jun 2021

HIV-1 Gag gene mutations, treatment response and drug resistance to Protease inhibitors: A systematic review and meta-analysis protocol

PONE-D-20-31760R1

Dear Dr. NKA,

We’re pleased to inform you that your manuscript has been judged scientifically suitable for publication and will be formally accepted for publication once it meets all outstanding technical requirements.

Kind regards,

Paul Spearman

Academic Editor

PLOS ONE
---

## [Editor Report · Acceptance letter]

18 Jun 2021

PONE-D-20-31760R1 

HIV-1 Gag gene mutations, treatment response and drug resistance to Protease inhibitors: A systematic review and meta-analysis protocol 

Dear Dr. Nka:

I'm pleased to inform you that your manuscript has been deemed suitable for publication in PLOS ONE. Congratulations! Your manuscript is now with our production department. 

Kind regards, 

on behalf of

Prof. Paul Spearman 

Academic Editor

PLOS ONE